

# The light's in my eyes: optical modeling demonstrates wind is more important than sea surface-reflected sunlight for foraging herons

Holly K.M. Brown[1], Margaret Rubega[1] and Heidi M. Dierssen[2]

[1] Department of Ecology and Evolutionary Biology, University of Connecticut, Storrs, CT, United States of America
[2] Department of Marine Sciences, University of Connecticut at Avery Point, Groton, CT, United States of America

## ABSTRACT

Multiple lineages of birds have independently evolved foraging strategies that involve catching aquatic prey by striking at them through the water's surface. Diurnal, visual predators that hunt across the air-water interface encounter several visual challenges, including sun glint, or reflection of sunlight by the water surface. Intense sun glint is common at the air-water interface, and it obscures visual cues from submerged prey. Visually-hunting, cross-media predators must therefore solve the problem of glint to hunt effectively. One obvious solution is to turn away from the sun, which would result in reduction of glint effects. However, turning too far will cast shadows over prey, causing them to flee. Therefore, we hypothesized that foraging herons would orient away from, but not directly opposite to the sun. Our ability to understand how predators achieve a solution to glint is limited by our ability to quantify the amount of glint that free-living predators are actually exposed to under different light conditions. Herons (*Ardea* spp.) are a good model system for answering questions about cross-media hunting because they are conspicuous, widely distributed, and forage throughout a variety of aquatic habitats, on a variety of submerged prey. To test our hypothesis, we employed radiative transfer modeling of water surface reflectance, drawn from optical oceanography, in a novel context to estimate the visual exposure to glint of free-living, actively foraging herons. We found evidence that *Ardea* spp. do not use body orientation to compensate for sun glint while foraging and therefore they must have some other, not yet understood, means of compensation, either anatomical or behavioral. Instead of facing away from the sun, herons tended to adjust their position to face into the wind at higher wind speeds. We suggest that radiative transfer modeling is a promising tool for elucidating the ecology and evolution of air-to-water foraging systems.

## INTRODUCTION

Birds have repeatedly, and independently, evolved foraging strategies that involve detecting prey in water, and striking at them through the air-water interface. To do so successfully,

Corresponding author
Holly K.M. Brown,
holly.k.brown@uconn.edu

they must contend with a number of visual challenges imposed by the optical properties of the water itself. First, water surfaces can reflect light. Sun glint (hereafter, "glint") refers to the reflection of sunlight by a water surface, directly toward the viewer (*Hochberg et al., 2011*; *Preisendorfer & Mobley, 1986*). Glint obscures detection of upwelling light from beneath the surface of the water, making it more difficult to see objects below the surface. Humans experience glint as visible bright spots on the surface of the water. Glare is the visual discomfort from the direct reflection of the sun light into their eyes (*e.g.*, *Signoroni et al., 2020*).

Cross-media predators, by definition, are attempting to locate submerged prey, and therefore we expect them to have evolved ways to compensate for glint. One obvious method of reducing effects of glint is to turn away: an animal that forages at the air-water interface should orient itself generally away from the sun under clear sky conditions if it aims to reduce visual exposure to glint. But by how much? The amount of glint exposure is a complex function of the viewing direction, field of view of the detector or eye, topography of a wind-blown sea surface, sun elevation and the spectral distribution of light.

Radiative transfer modeling, used in optical oceanography, has shown that glint is generally reduced with an increasing difference in bearing from the sun (*Mobley, 1999*). For example, assuming 5 m/s wind, a sun elevation of 60°, and a viewing angle looking 40° downward with respect to the horizon, the sea surface would reflect only about 3% of the skylight incident upon the sea surface for an animal viewing the sea surface facing directly opposite (180°) to the sun's bearing. This percentage remains fairly similar until the viewer is facing perpendicular (90°) to the sun's bearing, but begins to rise fairly quickly thereafter, to about 12% when facing directly into the sun's bearing (Fig. 7 in (*Mobley, 1999*). Under higher wind conditions and/or when the sun is directly overhead (sun elevation 90°), orientation plays less of a role in reducing glint (*Zhang et al., 2017*).

However, aquatic birds that hunt during the daytime must compensate for an additional challenge: they must be within striking distance of their prey without causing prey to flee. Even though glint is lowest at 180° to the sun's bearing, this is also the bearing that would cause a predator to hunt directly into its own shadow. Several aquatic prey species are known to flee when shadows pass overhead (*e.g.*, *Forward, 1977*; *Roberts, 1978*; *Yoshizawa & Jeffery, 2008*). Therefore, a bird hunting across the air-water interface on a sunny day is likely trading off its ability to see prey against the prey's ability to see them.

*Mobley (1999)* used radiative transfer modeling to show that when orienting a light detector at about 135° away from the bearing of the sun while measuring remote sensing reflectance of oceanic environments, the view of glint from the water surface is as low as possible over a wide range of water surface and environmental conditions, without facing directly into self-shadow. When the sun is overhead, however, orientation does not play a role in reducing glint. Mobley's work, by extension, suggests that although there is a range of orientations at which birds could reduce their exposure to glint, orienting at 135° to the bearing of the sun is the position in which birds hunting across the air-water interface can best trade off reducing glint exposure, while also avoiding casting shadows over potential prey. This logic assumes that birds see and perceive glint as we do, an assumption that may not be justified, given the limitations on what we know about avian vision. Nonetheless,

there is some limited, and anecdotal evidence that they do and that they may be trading off glint exposure against their own detectability as we predicted based on Mobley's (*1999*) radiative transfer modeling. For example, Brown Pelicans (*Pelecanus occidentalis*), were found to orient at an average of 135.6° to sun bearing (s.d. = 36.1°) as they dove for fish (*Carl, 1987*). We have also documented that a tern diving for fish in a pond, did so at about 140° to sun bearing, (documented on video three times in a row; it also did so several times in a row before the lead author started recording the behavior; Video S1 ). Even the behavior of non-aquatic avian predators suggests that they experience challenges from intense light conditions as we do. When the dark facial masks on Masked Shrikes (*Lanius nubicus*) were painted white, they oriented away from the sun to a greater degree than shrikes with black masks (*Yosef, Zduniak & Tryjanowski, 2012*) suggesting that sunlight reflecting from their facial feathers caused some visual discomfort.

Visual ecologists have demonstrated that orientation is important in visual function (*e.g.,* *Muheim, Phillips & Åkeesson, 2006*; *Penacchio et al., 2015*), but there are only a handful of studies that investigate orientation specifically with regard to hunting strategies (*Carl, 1987*; *King & LeBlanc, 1995*; *Yosef, Zduniak & Tryjanowski, 2012*; *Huveneers et al., 2015*). Orientation with respect to the sun may affect the ability to see prey, and therefore should be explicitly considered when studying the foraging ecology of visual, cross-media predators. Here, we examine the hypothesis that avian cross-media predators use body orientation to reduce glint in their strike zones while hunting, and that they do so in a manner that trades off glint exposure against self-shadow into their strike zones (Fig. 1). To test this hypothesis, we studied diurnal herons of the genus *Ardea*, which belong to a clade of piscivorous birds that have been specializing to hunt across the highly reflective air-water interface for over 50 million years (*Prum et al., 2015*).

We studied, specifically, two daytime-active herons, Great Blue Herons (*Ardea herodias*) and Great Egrets (*Ardea alba*; hereafter, "herons"). These species are good models for answering questions about cross-media hunting because they are numerous, conspicuous, widely distributed, and forage throughout a variety of aquatic habitats, on a variety of submerged prey. If herons were using orientation to trade off glint and self-shadow, we predicted that they would: a. orient in ways that minimize glint, and maximize the signal from upwelling light (*i.e.,* the light reflected by potential submerged prey items), as compared with what would be expected if heron orientation were random, and b. specifically, we expect them to orient at an average of 135° to the sun bearing. Great Blue Herons tend to be slightly more crepuscular than the more diurnal Great Egrets (*McNeil, Benoît & DesGranges, 1993*). In gathering data from both, our intent was to obtain generalizable information about how herons might compensate for glare while hunting through the air-water interface, over a wide range of daylight conditions. We employed radiative transfer modeling in a novel context to directly estimate the actual exposure of individual birds to glint, on the basis of their orientation to the sun, the sun elevation, and light conditions.

We also considered the hypothesis that heron body orientation could be related to wind. Orientation with respect to both sun position and wind direction have been widely recognized as important physiological mechanisms by which animals regulate body and

 

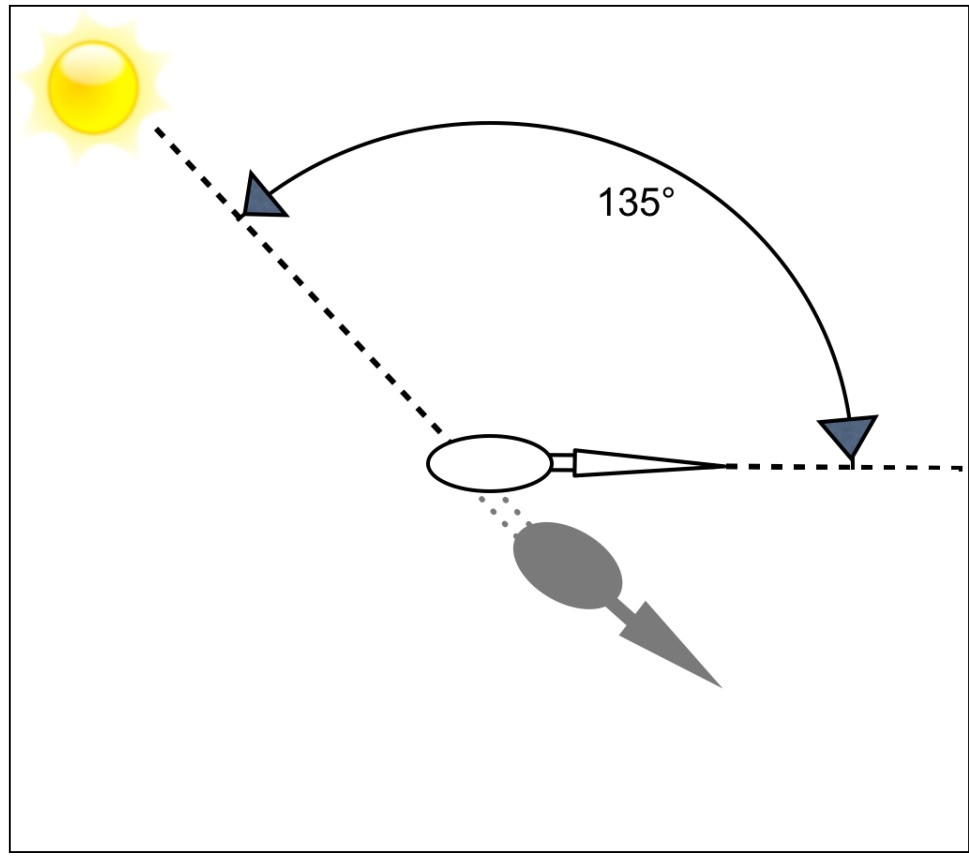

**Figure 1 Schematic representation of an overhead view of a foraging heron.** We predicted that herons would orient generally away from the sun to minimize visual exposure to glint; if they also avoid casting shadows over prey (which alert prey to their presence), we predict that they will orient at about 135° away from the bearing of the sun.

microhabitat temperatures (*e.g.*, *Orr, 1970*; *Walsberg, 1993*; *Fortin, Larochelle & Gauthier, 2000*; and many others). If herons were orienting to prevent loss of body heat, we predicted that heron orientation would correlate with wind direction, particularly at higher wind speeds. In addition, because of light reflecting from differently oriented wave facets, the advantages of orienting away from the sun are also reduced under higher wind speeds compared to flat conditions. Hence, we predicted that they would be more likely to face into the wind, regardless of sun position, as wind speed increased.

## MATERIALS & METHODS

In June 2013, January 2014, and January-February 2015, we opportunistically obtained 279 observations of 68 free-living, foraging herons in southern Florida (Fig. 2). As our study involved focal animals in the field, it was not possible to record data blind. In areas where an individual heron seemed to be actively guarding a foraging area (*e.g.*, by chasing other individuals away), we sampled the site only once. However, in areas where there were several herons foraging, we were able to obtain observations of different individuals
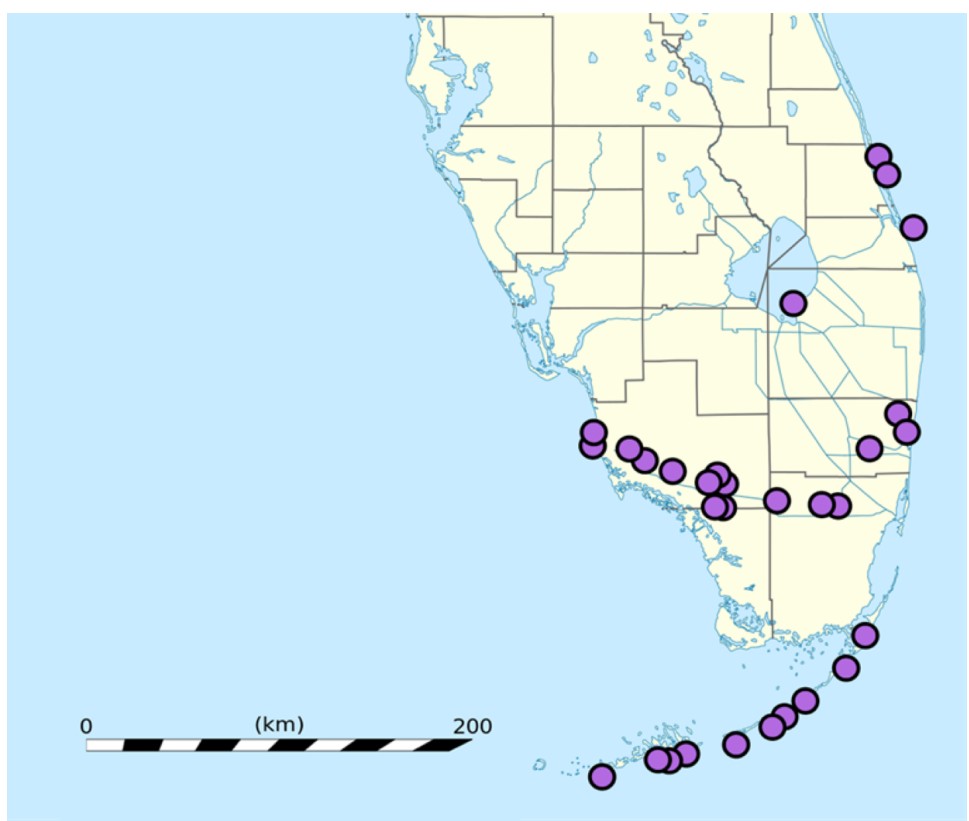

**Figure 2 Locations at which herons were observed in southern Florida, USA.** Circles indicate where data were collected. Some circles overlap more than one sampling location. Image Credit: The map was cropped from the original, "dministrative Map of Florida," created by Eric Gaba, *via* Wikimedia Commons (c) 2008 (user ID: Sting, CC BY 3.0).

at the same location. From a minimum distance of 20m, using binoculars (Nikon Monarch 3, 10x42), we observed individual foraging herons, and recorded up to six instantaneous samples of their body and head orientations, or fewer, if the individual flew away. In 2013, instantaneous samples were spaced 5 min apart, while in 2014 and 2015 samples were taken every two minutes. At the beginning of each observation, we noted the date, time, and the species. We noted sun visibility and cloud cover, as our predictions depended on the sun's being visible. We also noted wind bearing and speed category: calm/virtually undetectable (approximately 0 m/s); leaves rustle (approximately 5 m/s); branches sway (approximately 10 m/s); trees sway (approximately 15 m/s). If the wind speed and direction were noticeably variable, we updated this information during every instantaneous sample. We also noted whether the bird's shadow was obstructed (*e.g.*, by emergent vegetation) during every instantaneous sample. The orientation of the birds' bodies and heads, and the orientation of the wind were estimated in the field with a handheld compass. Exact sun bearing and elevation were later retrieved from the National Oceanographic and Atmospheric Administration's online sun position calculator, based on the time recorded for each instantaneous sample in each observation. Also, the approximate latitude and
longitude of each location were obtained from Google Earth. We were then able to calculate the estimated orientation differences between the birds and the sun, and between the birds and the wind, to use in analyses. After the final field season, we quantified error associated with estimating heron orientation. To do so, a field assistant placed a Great Blue Heron lawn ornament in 24 different directions and recorded its real orientation while the lead author (who made all compass measurements in the field) estimated each orientation from a distance of 20m, with the same binoculars that were used in the field (Nikon Monarch 3,10 × 42). Our estimations of body orientations of a Great Blue Heron lawn ornament were within an average of 9.3° (s.d. = 7.5°, N = 24). Therefore, we believe our estimates of heron body orientation are sufficiently accurate to test the hypotheses of interest in this study.

Using the *Hydrolight*® radiative transfer model (*Mobley, 1998*), we estimated absolute and relative glint in the green spectrum (550 nm). We defined relative green glint as green glint divided by the all incoming green light in the field of view (*i.e.,* water-leaving radiance plus radiance reflected by the sea surface). The relative measure is a type of signal-to-noise measurement. We used green light exclusively, because we wanted our estimates of glint to be conservative, in that they would maximize background brightness and minimize relative glint. In general, green light penetrates furthest into coastal waters (*Kirk, 2011*), and is therefore the light most available to be reflected by the seafloor, and exit the water as upwelling light. Therefore, in comparison, the contribution of glint is lower relative to the upwelling light in the green spectrum *versus* in other spectral regions.

We also used conservative but realistic values representing conditions in Florida, and conditions for wild foraging herons, for all variables in *Hydrolight*®, to obtain conservative estimates for glint. We modeled reflectance for a light-colored sand (ooid), which is both typical of many Florida coasts, but also creates high reflectance off of the sea floor, thus increasing the relative signal of upwelling light as compared with glint. We used a medium value for light attenuation in the water by indicating that light would attenuate by half for every meter below the sea surface (*McPherson et al., 2011*). We also used a water depth of 28 cm based on multiple records of the water depths in which herons forage in Florida (*Powell, 1981*; *Bancroft, Gawlik & Rutchey, 2002*). We ran the model under two wind conditions: 5 m/s, and 10 m/s. We also ran each of those models under two light conditions: where the sun was visible, and where it was obstructed by clouds.

To continue in our effort to gain conservative absolute and relative estimates of green glint, we also only retrieved outputs from a viewing direction of 40° to the nadir, because this viewing direction minimizes the proportion of skylight reflected at the sea surface under multiple wind speed scenarios (see Fig. 6 in *Mobley, 1999*).

We generated interpolated heat maps of absolute and relative glint in *Matlab (2015)*, using estimates based on sun elevations of 0°, 15°, 30°, 45°, 60°, 75° and 90° from the horizon. Finally, we used two-sample $t$-tests to compare estimates of visual exposure to glint for each heron body orientation with estimates of glint exposure that we would expect if heron orientation were random, using coordinates generated from random.org, to test the prediction that herons would orient in a manner that reduced visual exposure to glint.

All other statistical analyses were performed in R (*R Core Team, 2013*). To examine heron orientation relative to sun position, we used only instantaneous samples where herons' heads were oriented in the same direction as their bodies, and where the herons' shadows were cast over water (*i.e.,* unobstructed by emergent vegetation, and not cast onto land), and the sun was at least partially visible. To test the prediction that herons would orient 135° to the bearing of the sun, we regressed the absolute difference between sun bearing and heron bearing (*i.e.,* heron orientation relative to sun bearing), against sun elevation, using generalized estimating equations (*Liang & Zeger, 1986*) with the "geepack" package in R (*Højsgaard, Halekoh & Yan, 2006*). We used sun elevation as the independent variable instead of time, so that we were comparing heron orientations under consistent sun positions each day. To account for use of multiple observations for some individual herons, which are likely correlated, we used generalized estimating equations (GEE). GEE, an extension of generalized linear models, is a statistical approach for estimating regression parameters with clustered data (*Liang & Zeger, 1986*). We used "individuals" as grouping factors in our GEE model. We also used unstructured correlation matrices because we were unsure of what, if any, kind of relationship there might be among intra-individual data points. We then used the Wald-statistic to test the overall significance of the regression. As there is no package to estimate power or effect size based on a Wald test, at present, we estimated the effect size of our findings using a power analysis based on a generalized least squares linear model of our data using the "pwr" package (*Champely, 2015*). We similarly regressed the absolute value of the difference between wind and heron orientation *versus* wind speed category; and the absolute value of the difference between sun and heron orientation *versus* wind speed category as an ordered factor.

## RESULTS

Our prediction that herons would orient in a manner that suggested trading off reducing glint and self-shadow in their strike zones was not supported. We detected no departures from random orientation with respect to sun bearing in herons, across all sun elevations ($y = 89.37°$ $-0.02x$; $W = 0.003$; $p = 0.96$). This held true no matter whether we included the whole data set in the analysis, or just the subset of the data from when both the sun and the heron's shadows were visible (Figs. 3A–3B). The calculated effect size when regressing heron orientation relative to sun bearing *versus* sun elevation was miniscule ($[r^2 / (1 - r^2)]$ $= \sim0.0002$).

Overlaying our orientation data onto our heat maps displaying absolute and relative estimates of green glint added further evidence that herons are not using body orientation to reduce visual exposure to glint (Fig. 4). We decided to use only the heat maps we generated for 5 m/s wind speed for analyses because the heat maps generated for 10 m/s wind appeared nearly identical. There was no difference between the glint estimated at each heron orientation to the sun and glint that would be experienced at random, either in absolute ($t = 1.76$, $p = 0.08$) or relative estimates of green glint ($t = 1.55$, $p = 0.12$) with a visible sun. With a completely obstructed sun, glint was still no different from random for absolute ($t = 0.14$, $p = 0.89$), or relative estimates of green glint ($t = 1.10$, $p = 0.27$).
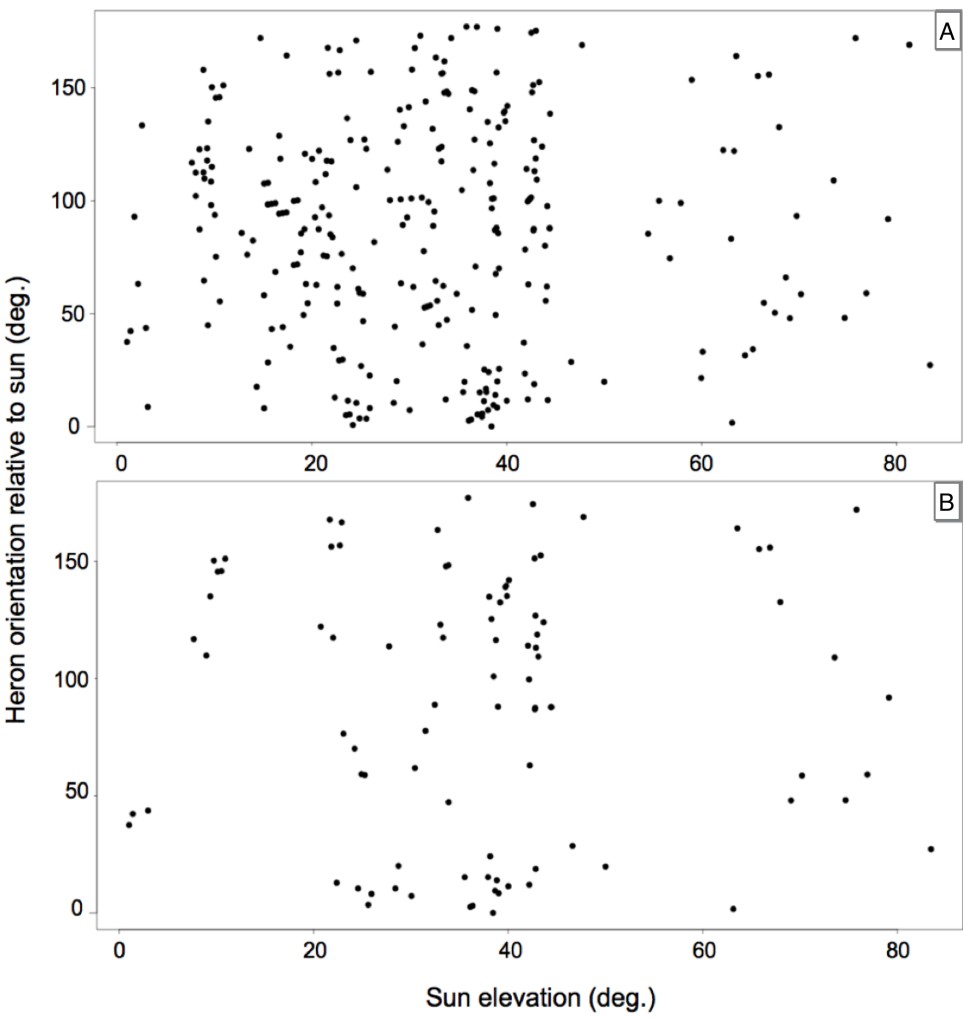

**Figure 3** **Scatterplots of our data, using points where heron's head and body orientations were in parallel.** In both graphs, 0° on the x-axis indicates that the sun is on the horizon, and 90° indicates that the sun is overhead. On the y-axis, 0° indicates that the heron was facing directly into the sun, and 180° indicates that the heron was facing opposite to the sun. (A) Using the whole data set, note that heron orientation with respect to sun position does not support our predictions, but rather appears random ($p = 0.95$). (B) Using only data points where the sun was unobstructed (*e.g.*, by clouds) and the heron's shadow was unobstructed (*e.g.*, by emergent vegetation) thins the data substantially, but does not change the overall lack of pattern ($p = 0.96$).

In support of our last hypothesis, herons tended to face more into the wind with increasing wind speed. With each increase in ordered categorical wind speed, herons faced about 31.70° (SE = 9.39°) further toward the sun ($W = 114$, $p = 0.0007$) (Fig. 5).

## DISCUSSION

Our data provide strong evidence that herons are not using body orientation with respect to sun position as a behavioral mechanism for reducing glint in their strike zone. Herons, when foraging through the highly reflective air-water interface, are not avoiding orienting

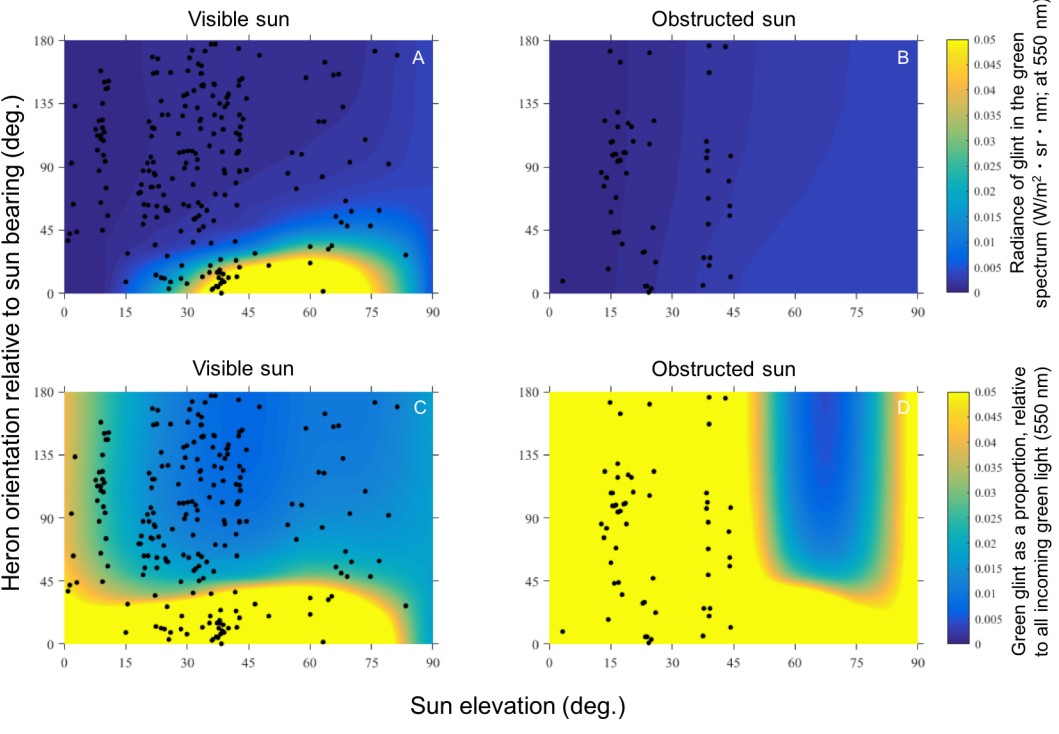

**Figure 4** **Heat maps that indicate measures of green sun glint (550 nm) directed at the viewer when wind is 5 m/s, by absolute (A, B) and relative (C, D) measures.** Our orientation data from Fig. 3B are superimposed on these heat maps by sun visibility: sun visible (left) and sun obstructed (right). Note that herons are orienting randomly, and they are foraging in "hot spots" where glint is maximized. (Heat maps for wind speeds of 10 m/s were nearly identical, and are not shown here.)

in directions with the highest estimated exposure to glint, whether in absolute terms, or relative to incoming light, and they are not trading off glint and shadow in their strike zones.

Our data also showed that as wind speed increased, herons tended to face more head-on into the wind. In retrospect, this is not surprising. One likely explanation for orienting to the wind is that herons are orienting for thermoregulatory purposes. Facing into the wind has been shown to decrease heat loss in birds (*e.g.*, *Fortin, Larochelle & Gauthier, 2000*).

Herons, and other cross-media hunters, may compensate for glint in other ways. *Krebs & Partridge (1973)* hypothesized that Great Blue Herons tilt their heads and long necks toward the sun –in a foraging behavior known as "head-tilting" (*Meyerriecks, 1962*) –to effectively shift a perceived area of glare out of their intended strike path. However, their idea was not fully tested, and further research is required to assess whether herons are head-tilting to compensate for glint effects, or has a different function. Relatedly, differences in head placement during foraging might account for some variability in heron body orientation with respect to sun elevation. Future research could investigate the direction of viewing, and eye movement in relationship to glint.

Because light that is reflected from water surfaces is polarized, some form of polarization vision could be useful to cross-media hunters. Theoretically, there are at least two ways to

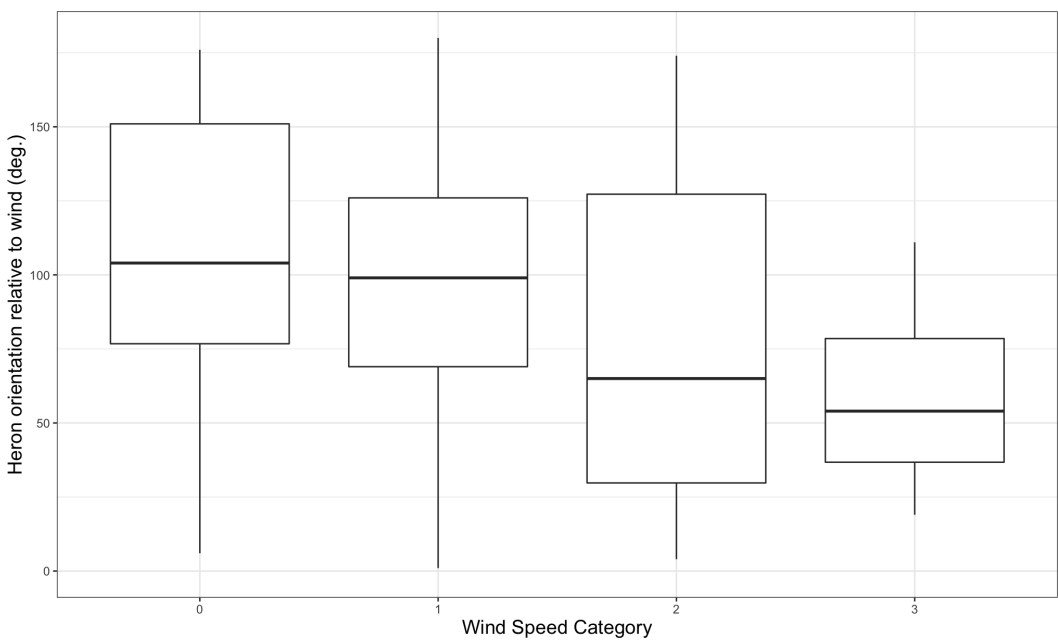

**Figure 5 Box plot of heron orientation with respect to the bearing of the wind by wind speed category (0 = calm or ∼0 m/s; 1 = leaves rustle or ∼5 m/s; 2 = branches sway or ∼10 m/s; 3 = trees sway or ∼15 m/s).** On the $y$-axis, 0° indicates that the heron was facing directly into the wind, and 180° indicates that it was facing opposite to the wind. With each increase in ordered categorical wind speed, herons faced about 31.70° (SE = 9.39°) further toward the sun ($W = 114$, $p = 0.0007$).

filter polarized light: before the image is projected onto the retina, or after. For polarized light to be filtered prior to reaching the retina, herons would need an anatomical feature capable of excluding it (as polarized sunglasses would). However, we are unaware of any evidence for such anatomical features in any natural visual systems. For polarized light to be filtered after reaching the retina, we would suggest some neurological capability that allows herons to detect, and subsequently ignore the polarized light as "noise", similar to how we would filter out a din of many voices while conversing with a friend in a crowded restaurant. This possibility seems much more likely. There is considerable evidence that animals across many taxa can see patterns of light polarization, including some behavioral evidence in birds (reviewed in *Muheim, 2011*). We suggest that behavioral studies designed to detect whether avian cross-media predators exhibit polarized light sensitivity would be fruitful for understanding the ability of these birds to hunt successfully in the face of intense glint.

Herons may be a good system in which to look for a mechanism for polarization vision in birds because they do not seem to be behaviorally compensating for glint. To date, we only have some very basic details about herons' visual anatomy. In Great Blue Herons, fine structures of the pecten (*Braekevelt, 1991*), rods and cones (*Braekevelt, 1994*) and cone pigments (*Braekevelt & Young, 1994*) have been studied. Their rod to cone ratio is also documented; it is 0.6:1, which puts them in between the diurnal herons (*e.g.*, the Tricolored Heron, *Egretta tricolor*, has a rod to cone ratio of 0.3:1) and the nocturnal

herons (*e.g.*, the Yellow-crowned Night Herons, *Nycticorax violaceus*, has a 2.3:1 rod to cone ratio; *Rojas et al., 1999*). Generally, higher rod to cone ratios produce better night vision than lower rod to cone ratios (*Rojas et al., 1999*). In Great Egrets the only scholarly reference we found regarding their eye anatomy dates back to a 1917 book, in which the basic macrostructure of the pecten, and the "dominant color" of the fundus were described (*Wood, 1917*).

Although we have suggested some avenues of study to investigate other potential behavioral or anatomical mechanisms for reducing glint exposure in herons, it is possible that there are cross-media predators that do use body orientation to mitigate effects of glint. There is some anecdotal evidence for this behavior in Brown Pelicans (*Pelecanus occidentalis*) and terns (*Carl, 1987*; Supporting Information). Therefore, we suggest further study of body orientation relative to sun bearing in these, and other, plunge-diving birds.

Lastly, we suggest continuing to use radiative transfer modeling to better understand the light conditions that air-to-water cross-media predators face while foraging. This new tool might also be useful for re-examining work from previous studies, for example on the physical conditions that affect foraging success in piscivorous birds (*e.g.*, *Grubb, 1977*; *Bovino & Burtt, 1979*; *Carl, 1987*). Understanding the conditions under which birds view their prey will lead to deeper understanding of their visual and behavioral ecology.

## CONCLUSIONS

In this study, we tested the hypothesis that foraging herons would orient away from the sun to avoid experiencing glare due to sunlight reflecting from surfaces of the water bodies in which they hunt. Field observations of heron body orientation, along with our estimations of sun glint *via* radiative transfer modeling provided evidence against our hypothesis; herons did not tend to orient in a manner that reduced their exposure to glint, but rather oriented to face the wind at higher wind speeds. Radiative transfer modeling, a tool from optical oceanography, was useful for investigating visual and behavioral ecology in this air-to-water foraging system, and should be considered in similar studies.

## ACKNOWLEDGEMENTS

We are grateful to the Smithsonian Marine Station, where a portion of these data was collected while HMB was working under the advisement of C.S. McKeon. For edits and helpful commentary, we are also grateful to T.W. Cronin, the entire Ornithology Group at the University of Connecticut, Andrew Moiseff, Eric Schultz, the late Eldridge Adams, and reviewers.

### Funding

This work was supported by Sigma Xi, via a Grant-in-Aid of Research (G20131015280936 to Holly Milton Brown), the University of Connecticut, via the Crandall Cordero Fellowship and the Department of Ecology and Evolutionary Biology zoology award (to Holly Milton

Brown), a Link Foundation/Smithsonian Institution Fellowship (to Holly Milton Brown), and National Aeronautics and Space Administration's Ocean Biology and Biochemistry grant (NNX15AC32G to Heidi M. Dierssen). The funders had no role in study design, data collection and analysis, decision to publish, or preparation of the manuscript.

**Grant Disclosures**

The following grant information was disclosed by the authors:
Grant-in-Aid of Research: G20131015280936.
University of Connecticut.
Link Foundation/Smithsonian Institution Fellowship.
National Aeronautics and Space Administration's Ocean Biology and Biochemistry grant: NNX15AC32G.

**Competing Interests**

The authors declare there are no competing interests.

**Author Contributions**

- Holly Milton Brown conceived and designed the experiments, performed the experiments, analyzed the data, prepared figures and/or tables, authored or reviewed drafts of the paper, and approved the final draft.
- Margaret Rubega conceived and designed the experiments, authored or reviewed drafts of the paper, and approved the final draft.
- Heidi Dierssen conceived and designed the experiments, analyzed the data, prepared figures and/or tables, authored or reviewed drafts of the paper, and approved the final draft.

**Data Availability**

Raw data and code is available at Figshare:

Brown, Holly (2021): Body Orientation Data. figshare. Dataset. https://doi.org/10.6084/m9.figshare.5675125.v1.

Brown, Holly (2021): R and MATLAB codes. figshare. Journal contribution. https://doi.org/10.6084/m9.figshare.5675134.v1.

Brown, Holly (2017): Tern diving at 140 deg. to sun bearing. figshare. Media. https://doi.org/10.6084/m9.figshare.5675092.v1.

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
