# Peer review of "The light’s in my eyes: optical modeling demonstrates wind is more important than sea surface-reflected sunlight for foraging herons"

_PeerJ, doi:10.7717/peerj.12006_

## Round 0.1 · original submission · Minor Revisions

I apologize that it took a little longer than usual to reach a decision on your MS. In the end, I was fortunate to receive two expert reviews for your MS and both reviewers provided positive feedback on your work. However, both had some comments on the literature reviewed and the statistics presented that you will need to respond to before I can accept your MS. Reviewer 1 also had a suggestion concerning your title and focus and I think it is up to you to decide if you want to make changes to reflect this suggestion. I do agree that the title should be more informative in terms of your actual findings. I agree with the reviewers that you have addressed an important question and that your rationale and methods are straightforward. I have only a few minor suggestions of my own.

I would caution you to avoid phrases such as “in order to” because you cannot determine the cause of behaviors such as orienting from the sun (e.g., line 29). You should rephrase to say that they orient away from the sun, which would result in reduction of glint effects.

Avoid using “since” and “while” in non-temporal contexts (e.g., lines 33, 48, 78). Replace with “because” and “although or whereas.”

Avoid single sentence paragraphs (e.g., lines 73-76). Each paragraph should have a thesis statement, supporting body and concluding sentence.

You can delete “or not” on line 150.

·

Basic reporting

The manuscript was a joy to read. Very clear and unambiguous language throughout.

The first paragraph of the introduction is a bit thin on references while describing some things that I am sure there is established literature on? Perhaps some references to the properties of glint would be useful here. At least for the uninitiated, especially as we’re told that it’s wrong to call it glare. What’s the difference, and what do you base that claim on?

Otherwise, the introduction provides very clear and useful background for this work, though perhaps “James A. Rogers (1983) Foraging Behaviour of Seven Species of Herons in Tampa Bay, Florida, Colonial Waterbirds, 6, pp11-23)” ought to be included in the introduction?

The article structure is well presented and clear. Raw data are easily accessible and the code provided is clear. The manuscript is also self contained and presents clearly and unambiguously what the hypothesis, methods, results, and conclusions are.

I would suggest a different title. While the use of "quantitative optical models/radiative transfer modeling" looks to be novel and was really useful in showing what level of glint the birds were likely to experience, the story that I find most exciting is that they seem to not be bothered by glint. I think a title focusing on that would bring this work to the attention of an audience interested in bird vision.

Experimental design

This work is original as far as I can tell, which is odd as it seems such an obvious thing to investigate. Definitely relevant and meaningful. I am impressed with the clear hypothesis and focused investigation put into testing the hypothesis.

This study is based on a lot of field work collected over a long period of time. The data are purely observational, so there are little to no ethical concerns. Efforts were made to account for observer error, which is quite important in this kind of work and modern technology was used to obtain very detailed information on things that are not easily discerned in the field (e.g. elevation of the sun). A tool to measure actual strength of light on site and perhaps images of water surface for later analysis of wind disturbance might have been useful, but considering the results, it’s unlikely that these things would have changed the outcome much.

The methods are clearly described and could be replicated.

Validity of the findings

While not an expert on the statistics used, the data are robust and the statistical analyses reflect what is fairly clear from the very first plot of the data. I wonder if it is possible to use ordinal analyses on the wind/orientation data as there seems to be a trend towards stronger wind having stronger effect as it increases in strength? When the model is not informed of a trend in the wind classifications, it simply finds that the strongest wind is different from the others. Am I interpreting that correctly?

A fairly large section of the discussion focuses on polarised light as a method of seeing through the glint. This was also my first thought (and google search) and as such it points out a rather glaring (pardon the pun) hole in the literature with regards to exactly how birds see through the surface of the water, when it seems glint does not bother all of them. The authors suggest that herons be used as model animals to investigate this and that makes sense to me.

Additional comments

I was delighted to review this manuscript. It presents such a clear hypothesis and tests it so thoroughly. I'm no expert on bird vision, so I can't really speculate on whether your findings were to be expected or not. They certainly surprised me, especially considering the scarcity of data suggesting birds are better able to see through glint than e.g. humans.

Reviewer 2 ·

Basic reporting

This manuscript asks and answers an interesting question: do herons orient their body relative to the sun to optimize viewing conditions while hunting?

General Comments:
Strike angle: In the introduction, the authors mention a viewing angle looking 40 degrees downward. Is there any information on the strike angle taken by herons? For example, do they strike at 40 degrees (which is more out than down), or do they strike more perpendicular to the water’s surface (70-90 degrees)? And what effect would that have on glint? From personal experience, it is easier to see into water from directly above than to see into water out in front of me. Perhaps herons don’t orient away from glint because they are looking down, not down and away.

Minor Comments:
Line 20 – I would not go so far as to say ‘ubiquitous’.
Line 21 – Change ‘signals’ to ‘cues’. A signal means that both parties receive a benefit or that information is intentionally being displayed by the sender.
Line 20-22 – This sentence is generally difficult to read with so many commas.
Line 26 and 32 – the ‘spp.’ should not be italicized since it is not Latin.
Line 28-29 – Earlier, the authors implied that this strategy would not work well. I suggest they remove the earlier sentence and turn this sentence into a proper hypothesis and prediction.
Line 44 – The ‘hereafter’ is unnecessary since the authors already called it ‘glint’ and define it.
Line 48 – Have the authors consider what is already known about how archer fish do the same thing but in the opposite direction? Temple et al. 2010 in PRSLB, Ben-Simon et al. 2012 in J of Vision and in J of Exp Bio, among others
Line 66 – additional and opposing to what other challenge?
Line 67 – the order in which information is revealed makes this section difficult to read… maybe just get rid of lines 65-66 since you talk about the trade-off later anyway.

Experimental design

The methods are carefully chosen and well executed.

General Comments:
Statistics: I have not used GEE statistics before, but I understand that it is good for avoiding pseudoreplication in longitudinal studies (repeated measures of an individual over time), which is exactly what the authors have here. So the statistics used are appropriate. My question though is regarding the data in Figure 5. The authors compare the wind speed categories two at a time. But wind speed is ordinal, and therefore, it would be valuable to have a test that determines significance (or not) of all four categories together. There appears to be an overall trend, but is that trend significant, as a whole? Wind speed, in truth, is continuous not categorical. I would like to see a comparison made where the step-wise relationship between wind speed categories is recognized.

Line 245 – How are there three estimates of alpha error, but only one estimate of beta error? Also power (1-beta error) is irrelevant if the result is significant because the null hypothesis has been rejected, so there is no chance of committing Type II error (incorrectly accepting the null).

Validity of the findings

The conclusions are interesting and useful.

Comments:

Line 269-275 – The logic here needs some work. Being able to detect polarization is not the same as filtering polarized light. Quite the opposite in fact. If herons were able to see and detect polarized light, they would be guaranteed to fall victim to glint. If herons were not able to detect polarized light, it wouldn’t mean anything one way or the other.

---

## Round 0.2 · accepted · Accept

Thank you for attending to the reviewers' requests for minimal revision. I think this is a clearly written, straightforward paper that is now ready for publication.